# Characterisation of Macrophage Polarisation in Mice Infected with Ninoa Strain of *Trypanosoma cruzi*

**DOI:** 10.3390/pathogens10111444

**Published:** 2021-11-06

**Authors:** Dunia M. Medina-Buelvas, Miriam Rodríguez-Sosa, Libia Vega

**Affiliations:** 1Department of Toxicology, Centre for Research and Advanced Studies of the National Polytechnic Institute, Av. IPN 2508, Zacatenco, GA Madero, Ciudad de México 07360, Mexico; duniamed@gmail.com; 2Unidad de Biomedicina, Facultad de Estudios Superiores Iztacala, Av. de los Barrios 1, Los Reyes Iztacala, Tlalnepantla 54090, Estado de Mexico, Mexico

**Keywords:** macrophage polarisation, *Trypanosoma cruzi*, chagas disease, ninoa strain, parasitic burden, BALB/c mice, C57BL/6 mice

## Abstract

Macrophages (MΦ) play a key role in the development of the protective immune response against *Trypanosoma cruzi* infection. To determine the role of MΦ subtypes M1 and M2 in the development of immunity against the Mexican strain of *T. cruzi* (Ninoa strain), we have analysed in a time course the infection and characterised the M1 and M2 subtypes in two mouse models, BALB/c and C57BL/6. After infection, BALB/c mice developed an increased blood parasite load and the parasites were cleared from the blood one week later than in C57BL/6 mice. However, similar cellular infiltrate and cardiac alterations were observed between BALB/c and C57BL/6 mice. At 36 days, the *T. cruzi* infection differentially modulated the expression of immune cells, and both the BALB/c and C57BL/6 mice significantly reduced TCD4+ cells. However, BALB/c mice produced significantly more TCD8+ than C57BL/6 mice in the spleen and lymph nodes. Furthermore, BALB/c mice produce significantly more MΦ in the spleen, while C57BL/6 produce similar levels to uninfected mice. The M1 MΦ ratio increased significantly at 3–5 days post-infection (dpi), but then decreased slightly. On the contrary, the M2 MΦ were low at the beginning of the infection, but the proportion of M1 and M2 MΦ at 36 dpi was similar. Importantly, the MΦ subtypes M2c and M2d significantly increased the induction of tissue repair by the end of the acute phase of the infection. These results indicate that the Ninoa strain has developed strategies to modulate the immune response, with fine differences depending on the genetic background of the host.

## 1. Introduction

Chagas disease or American trypanosomiasis is a neglected tropical disease historically endemic to continental Latin America, that already affects 6 to 7 million persons worldwide [1,2], with 30,000 new cases reported annually for all forms of transmission. Over 70 million persons are at risk of developing the infection, with 8000 newborns infected during gestation and around 12,000 persons dying every year [3,4]. Chagas disease is a parasitic, systemic, and chronic infection, which is caused by the hemoflagellate protozoan *Trypanosoma cruzi* (*T. cruzi*). Vectorial transmission is the most recognised, and it is through the bite site or by contact with the faeces and/or urine of around 150 species of hematophagous triatomine bugs or “kissing bugs” contaminated with the parasite. Nevertheless, multiple forms of non-vectorial ways have been reported, such as congenital (mother to foetus during pregnancy), oral, blood transfusion, tissue transplantation, or laboratory accidents [5], that are relevant and responsible for the spreading from endemic to non-endemic areas. 

Once *T. cruzi* parasites enter the circulation they are internalised by blood cells such as monocytes and macrophages (MΦ) in which the parasites proliferate [6]. After 60 days of the initial infection, the parasites are cleared from the blood, followed by a long-term clinical condition that can persist for 2–3 decades, referred as the chronic phase (in 20–30% of infected individuals). This stage has a very low parasite load only detected in specific organs, and whose severity depends on the time since infection, host immune status or genetic background [7]. It is characterised by parasites remaining within the muscles and the myocardium that evoke a chronic immune reaction related to death by a long-term heart condition called chronic chagasic cardiomyopathy [8], or severe gastrointestinal complications characterised by megacolon and/or megaesophagus [9]. 

For the study of the pathology, as well as the immune response, associated with *T. cruzi* infection, several murine models of the infection have been developed [10]. Currently the murine model of the *T. cruzi* infection has been well characterized; for example, it is known that independent of the *T. cruzi* strain, the male are more susceptible to infection than female mice [2,11]. Moreover, the mice frequently have been used as a functional model of Th1 and Th2 cell responses. Therefore, control of *T. cruzi* infection in resistant mice such as C3H and C57BL/6, is associated with the activation of components of the innate immune system, including MΦ and natural killer (NK) cells [12,13,14], inflammatory cytokines as the macrophage migration inhibitory factor (MIF), and the tumour necrosis factor (TNF)α, as a Th1-type response associated to interferon (IFN)γ [15,16]. In contrast, susceptible BALB/c mice develop cardiomyopathy or severe gastrointestinal complications after *T. cruzi* infection and mount a Th2-type response that is associated with the production of the cytokines interleukin (IL)-4 and IL-10 [17,18].

Furthermore, recent studies have shown that MΦ polarisation to the M1 or M2 subtype also plays a critical role in the control or the severity of *T. cruzi* infection [19]. The type of continuous stimuli to which MΦ are subjected determine the M1 and M2 states. Thus, the Th1-inflammatory stimulus, IFNγ + lipopolysaccharide (LPS) or TNFα, generates the M1 type. While M2 MΦ has subtypes depending on the stimulus to which they are subjected; IL-4 generates M2a, the activation induced by Fc receptors and immune complexes generate M2b, the IL-10 and glucocorticoids generate M2c [20], and the M2d subtype results from adenosine-dependent “switching” of M1 [21]. This careful categorisation also made distinctions between M2 MΦ functions; M2a play a role in parasite death and encapsulation of extracellular parasites, M2b are involved in Th2 activation, M2c participates in matrix deposition and tissue remodeling, and M2d induces angiogenesis [22,23].

A recent study reported that susceptible BALB/c mice treated with concanavalin-A favours peritoneal M1 MΦ polarisation to increase trypanocide activity, resulting in ameliorated systemic *T. cruzi* infection [24]. Although these results demonstrate that M1 induce protection against *T. cruzi* infection in susceptible BALB/c mice, other studies indicate that M2 regulates the lethal nitric oxide (NO) release during *T. cruzi* infection, along with anti-inflammatory cytokines produced by Th2 cells, such as IL-10 that promotes protection from fatal acute myocarditis [25]. Thus, the balance of a pro-inflammatory-M1 and anti-inflammatory-M2 response is essential to control parasite replication and prevent fatal myocarditis, respectively.

In order to provide new insights regarding the role of MΦ polarisation and their effects on infection, we compared the clinical and pathological findings, the percentage of CD4+ and CD8+ lymphocytes and the proportion of NK cells from the spleen and the lymph nodes using the Mexican *T. cruzi* Ninoa strain in two experimental models, C57BL/6 and BALB/c mice. We also evaluated the polarization of peritoneal M1 and M2 MΦ, together with the M2 subtypes in BALB/c mice.

## 2. Results

### 2.1. Parasitemia Burden in Blood of BALB/c and C57BL/6 Mice

To examine whether the genetic immunological background of mice, Th1-C57BL/6 vs. Th2-BALB/c, plays a relevant role in the blood burden of *T. cruzi* infection, we compared the parasite burden in blood from BALB/c and C57BL/6 mice. 

The presence of parasites was observed at day 8 post-infection for both strains. A rapid and continuous increase in parasitaemia for both mouse strains was observed, although in the BALB/c mice, the blood stream parasites reached a peak at day 20, while in the C57BL/6 mice, the blood stream parasites reached a peak at day 24 post-infection. Importantly, the infected BALB/c mice showed significantly higher levels of blood parasitaemia at 12, 28 and 32 days post-infection (dpi) than the C57BL/6 mice (* *p* < 0.05). Moreover, the BALB/c mice cleared blood parasites 4 days later than the C57BL/6 mice, on 36 and 32 days, respectively (Figure 1). Importantly, no mortality was observed in the BALB/c or C57BL/6 infected mice. 

### 2.2. Systemic Changes Due to T. cruzi Infection

There were no differences in the food and water consumption between the infected and non-infected mice in any of the experimental groups (data not shown). 

At day 36, after the clearance of blood parasites, the mice were euthanised and the weight of different organs was recorded. There was an increase in weight and size of the spleen, liver, and lymph nodes of the infected mice compared to the non-infected mice, however, no significant differences were observed between the infected BALB/c and C57BL/6 mice (Table 1).

### 2.3. Histological Findings in the Heart

At 36 dpi, the hearts from the infected BALB/c mice showed moderate inflammatory mononuclear cell infiltration but without major histopathological signs of lesions (Figure 2A,B). Importantly, after 180 dpi (6 months), the histopathology sections showed a reduction in the inflammatory reaction; however, signs of injury associated with an altered cardiac fibre architecture were observed (Figure 2C,D). The alteration of cardiac architecture was confirmed by analysing collagen deposits. The presence of collagen deposits was observed at 36 dpi (Figure 3A,B), and these notably increased after 6 months post-infection (Figure 3C,D). Similar histological findings in the heart from C57BL/6 mice were observed (data not shown).

### 2.4. Cell Populations in Spleen and Lymph Nodes

The proportion of immune cells, CD4+ and CD8+ T cells, NK (CD335+) cells and MΦ (F4/80+) in the spleen and lymph nodes was analysed at 36 dpi in both BALB/c and C56BL/6 mice. The data were acquired by flow cytometry and analysed by ascertaining the variation of the percentage of positive cells. 

In BALB/c, the infection by *T. cruzi* Ninoa strain significantly reduced the proportion of TCD4+ cells and increased the proportion of TCD8+ cells in the spleen and lymph nodes (*p* < 0.05). Moreover, there was an increase in the number of MΦ in the spleen (*p* < 0.05), but a reduction in their number in the lymph nodes (*p* < 0.05). The number of NKs cells was elevated in the lymph nodes; all were compared to the non-infected mice (Figure 4A,B). 

In C57BL/6 mice, the TCD4+ proportion was also significantly reduced in the spleen and lymph nodes (*p* < 0.05). There was no change in the proportion of TCD8+ or in NK cells in the spleen and lymph nodes. There was no change in the proportion of MΦ in the spleen, but there was a significant decrease in the lymph nodes (*p* < 0.05); all were compared to the non-infected mice (Figure 4C,D). 

### 2.5. Characterisation of Peritoneal Exudate Cells (PECs) after T. cruzi Infection

Because a significant change in the proportion of MΦ was observed between BALB/c and C57BL/6 mice, and because MΦ are key cells in the process of tissue repair, regeneration, and fibrosis induction [26], the intacellular cytokines typically produced by MΦ such as IL-10, TGFβ, TNFα and the vascular endothelial growth factor (VEGF) were analised in peritoneal MΦ (recognised as adherent PECs F4/80+) at 36 dpi in the BALB/c and C56BL/6 mice. The data were acquired by flow cytometry and were analyzed determining the variation in the percentage of positive cells.

The number of PECs from both BALB/c and C57BL/6 infected mice was three times higher than those obtained from non-infected mice. Peritoneal MΦ (F4/80+ PECs) from infected mice produced significantly more TNFα and VEGF (*p* < 0.05), and they presented a significantly reduction of TGFβ (*p* < 0.05) and similar levels of IL-10 in both BALB/c and C57BL/6 mice compared to the non-infected mice (Figure 5A,B, respectively).

Interestingly, F4/80-negative PECs (other cell types, not MΦ) produced significantly less IL-10, TGFβ, TNFα and VEGF (*p* < 0.05) compared to the non-infected mice as peritoneal MΦ from both BALB/c and C57BL/6 mice (Appendix A).

### 2.6. Macrophages Polarisation Induced by T. cruzi Infection

Because the production of pro-inflammatory and anti-inflammatory cytokines from PECs comes mainly from MΦ, the M1 and M2 subpopulations were characterized. Thus, to establish the kinetics of MΦ polarisation to M1 and M2 during the infection by *T. cruzi* Ninoa strain, peritoneal MΦ from uninfected (day 0) and infected BALB/c mice were evaluated over the time of the early phase of infection (3, 5, 12, 22 and 36 dpi). Because it has been reported that C57BL/6 mice hardly develop M2 MΦ in other parasitic infections [27,28] and based on our experience, we gave priority to the analysis of M2 in infected BALB/c mice. 

Non-infected mice showed low proportions of M1 and M2 MΦ with no significant difference. After infection, the proportion of M1 MΦ increased significantly at 3 and 5 dpi. Importantly, at 12 and 22 dpi, the proportion of M1 decreases, although it remains significantly higher compared to non-infected mice (*p* < 0.05). M2 MΦ decreased significantly from 3 to 22 dpi (*p* < 0.05); however, at 36 dpi, they increased significantly compared to non-infected mice (*p* < 0.05). It is important to note that the proportion of M1 and M2 at 36 dpi was similar (Figure 6A).

Next, we assessed whether different M2 MΦ phenotypes could play a role against early *T. cruzi* infection. Thus, the M2 MΦ subtypes from peritoneal MΦ were analysed over the time of the early phase of *T. cruzi* infection (3, 5, 12, 22 and 36 dpi) using non-infected mice as controls. 

At 3 dpi, only the M2c increased significantly (*p* < 0.05). At 5 dpi, the four subtypes increased significantly, M2a, M2b, M2c, and M2d (*p* < 0.05). At 12 dpi, M2b, M2c and M2d continued to increase (*p* < 0.05). At 22 dpi, only the M2d subtype was increased. Interestingly, at 36 dpi M2a, M2b, M2c, and M2d presented a higher increase than that at earlier infection times (* *p* < 0.05) (Figure 6B). These results indicate that the M2 MΦ change their subtype polarisation with time, even when they are not largely modified in relative numbers when compared to the amount of M1 MΦ. This result indicates that the plasticity of M2 MΦ is important in the control of the initial phase of *T. cruzi* infection.

Finally, intracellular production of IL-10, TGFβ, TNFα, and VEGF from peritoneal MΦ was also analysed at 0, 3, 5, 12, 22, and 36 dpi in BALB/c mice. Cytokine-producing peritoneal MΦ were not detected in non-infected mice. After infection by *T. cruzi*, peritoneal MΦ produced IL-10, TGFβ, TNFα, and VEGF at all of the times evaluated post-infection. Importantly, a higher proportion of MΦ producing TGFβ and TNFα was observed at 3 and 5 dpi (orange and grey bars, * *p* < 0.05), while the production of VEGF was increased at 3, 5, and 22 dpi (grey, yellow, and blue bars, * *p* < 0.05). 

## 3. Discussion

It is widely accepted that both the *T. cruzi* parasite strain and host genetic factors are important for disease progression. Thus, we used two host lines, BALB/c and C57BL/6 infected with strains of Ninoa *T. cruzi* to evaluate aspects of the acute phase of the infection and to determine the participation of the genetic background in the immune response in the course of infection.

Here, we show that the blood parasite burden in C57BL/6 mice was significantly lower at 12, 28, and 32 dpi, the maximum parasite peak had 4 days delay, and the parasites were undetectable 4 days earlier compared to similarly infected BALB/c mice. Thus, the C57BL/6 mice were more resistant to the infection and solved it earlier than the BALB/c mice. 

Even, when there were differences in the blood parasite burden between BALB/c and C57BL/6, no mortality or evident pathological differences in body weight, spleen, liver, lymph nodes, or heart were observed between the infected BALB/c and C57BL/6 mice. This is in accordance with previous reports on Ninoa strain, describing that even when parasitaemia is evident, there are no clinical manifestations and mortality is very low independent of the genetic background of the mouse [29]. Here we observed absence of mortality in both BALB/c and C57BL/6 mice, this may be due to the low virulence of the strain, the low number of parasites used in this study in comparison to other studies, where the inoculum went up to 10–13 times higher, and the use of females, which are less susceptible to *T. cruzi* infection than male mice.

Importantly, persistent infiltrated cells and collagen deposits, but no amastigote nests were observed in neither the BALB/c nor the C57BL/6 mice, although previous reports indicate that the Ninoa strain is cardiotropic [30]. This could probably be due to the low number of inoculated parasites. These results indicate that the damage to the heart is persistent even when no parasite nests were observed, and even when the parasitic burden is low, and the host appears to be fully recovered from the acute phase of the infection.

The small differences in blood parasite behaviour and heart damage between C57BL/6 and BALB/c mice could be explained by the fact that different strains of *T. cruzi* have different infectivity potentials [31]. In this sense, the Tc1 genotype, to which the Ninoa strain belongs, has a high intraspecific genetic and phenotypic diversity [32], which is determinant to recognize the surface molecules of the parasite by immune cells [33]. These characteristics could explain how the Ninoa strain exhibits more modulating behaviours of the immune response compared to other strains of *T. cruzi* [32]. Thus, it is possible that the Ninoa strain could manipulate immune cells to produce a tolerogenic cytokines profile regardless of the genetic background of the mouse.

The analysis at 36 dpi of the proportion of some immune cells showed no difference in the regulation of TD4+ cells between BALB/c and C57BL/6 mice, in both cases they decreased. This behaviour was similar that previously reported by Espinoza et al. which a significant decrease in CD4+ T cell activation was observed in the acute phase of infection with *T. cruzi* strain Ninoa [30]. Whereas the proportion of TCD8 + cells in spleen and lump nodes was significantly increased in the BALB/c mice but not in the C57BL/6 mice. Therefore, the BALB/c mice developed a mild inflammatory process characterised by a slight increase in the expression of TCD8+, NK CD335+, and with a considerable increase in MΦ F4/80+ in the spleen, whereas the C57BL/6 mice did not. This supports the idea that Ninoa strain differentially modulates the immune response (even with small differences) depending on the genetic background of the Th1 or Th2 mouse [32]. 

Many studies show the mechanisms by which *T. cruzi* modulates the immune response, however the immune response modulation by the Ninoa strain has not been adequately investigated. Herein the major differences were observed in MΦ F4/80+, the ratio of these cells increased significantly in the spleen of the BALB/c mice but not in the C57BL/6 mice. And in the lymph nodes it decreased significantly in the BALB/c and C57BL/6 mice. It is possible that the modulation in TCD8+ lymphocytes may be influenced by the production of cytokines by antigen-presenting cells such as DCs and MΦ depending on the mouse lineage. Consistent with this Barbosa et al. described that the Mexican strains of *T. cruzi* infect and modulate MHC-II, toll-like receptors, and cytokines production in DCs obtained from C57BL/6 and BALB/c mice differently, suggesting that these strains have developed particular modulator strategies to disrupt DCs, and consequently, host immune responses [32].

In line with what we mentioned above, here was observed that IL-10-producing PECs were significantly increased in the BALB/c mice but not in the C57BL/6 mice, this observation is in agreement with the previous report that demonstrated an increase in IL-10 serum levels during the acute phase, which decays and rises gradually during the course of *T. cruzi* Ninoa infection [30]. Ferreira et al. showed an early release of some sera cytokines of infected BALB/c mice, which did not occur in C57BL/6 mice [34]. The induction of the anti-inflammatory cytokine IL-10 in the BALB/c mice and its absence in the C57BL/6 mice agrees with the genetic background of the mice anti-inflammatory-Th2 profile of the BALB/c mice, and inflammatory-Th1 profile of the C57BL/6 mice.

It is interesting that the production of TNFα did not negatively affect the production of IL-10 in the PECs of the BALB/c mice. Huynh et al. referred that the presence of TNFα does not negatively regulate the production of IL-10 [35], on the contrary, TNFα can induce IL-10 production [36]. However, to our knowledge, this finding was not followed up and the regulation of autocrine IL-10 production induced by TNFα and subsequent signaling, gene induction, and function has not been investigated. Undoubtedly, the infection by *T. cruzi* Ninoa is a good model to deepen in the future in this field. On the other hand, this mixed cytokines profile could also indicate that the activation of the cells and the polarisation of MΦ is not complete and different subtype of cells are coexisting.

We also show here that not all cytokine-producing cells are influenced by the mouse lineage, for example, although TGF-producing PECs were negatively modulated and the proportion of TNFα- and VEGF-producing PECs increased, the changes observed on these cells were similar, and no differences were observed between the BALB/c and C57BL/6 mice.

Currently, it is widely accepted that *T. cruzi* is recognised by host defence cells, such DCs and MΦ, one of the preferred target cells of the infective forms of *T. cruzi* [37]. MΦ are cells with versatility and plasticity to reprogram their phenotypes in a wide range of functional states (e.g., M1 to various M2 subtypes), depending on the environmental stimulus and physiological functions [38]. 

The polarisation of MΦ towards M1 confer them the microbicidal properties associated with a production of pro-inflammatory factors, such as the TNFα, IL-1β, IL-6, IL-12, IL-18, IL-23, and IFNγ, and the release of reactive oxygen species (ROS) and NO [38]. On the contrary, the MΦ polarisation towards an alternative phenotype M2 is involved with resolving inflammation, restablishment of homeostasis, angiogenesis, and tissue healing, typically through arginase/ornithine/urea, the epidermal growth factor (EGF), VEGF, TGFβ, and the mannose receptor CD206 [39].

Thus, the interaction of *T. cruzi* with monocytes and MΦ can activate them to acquire pro-inflammatory functions important for the control of an infection during the acute phase that are thus involved in the outcome of the disease [39,40]. Therefore, the MΦ subtypes characterisation is essential to understand the role of MΦ in *T. cruzi* infection; however, few studies have tried to elucidate the profile of MΦ in this disease [41,42]. Here, we show that at the beginning of the infection, the BALB/c mice increased the proportion of peritoneal M1 MΦ responsible for the increased levels of TNFα detected at this time. This observation is in accordance with Rojas-Marquez et al., as they showed that at the beginning of the infection, the mice generate an intense inflammatory response, characterised by the presence of peritoneal and splenic MΦ M1 responsible for the increased levels of TNFα, IL-6, IL-1β, and NO in the acute phase of infection [43]. 

On the contrary, at the beginning of the infection, a significant reduction of M2 MΦ up to 22 dpi was observed, which increased at 36 dpi. However, the presence of IL-10 and TGFβ, and the later increase of VEGF suggests that there are M2 MΦ involved in controlling the acute phase of infection. These results are in line with previous reports that point out that the activation of the mTOR pathway and the presence of IL-10 in the acute phase of the infection suggest the presence of M2 MΦ, but later than M1 MΦ [43]. Thus, it is possible that at some point in the acute phase of *T. cruzi* infection, the M1 and M2 profiles co-exist in the host [41].

Finally, in order to delve into the existence of M2 MΦ during the acute phase of T cruzi infection, we characterised the subtypes of M2 MΦ in infected BALB/c mice. We found at 5 dpi, a significant increase in M2b, M2c, and M2d MΦ that prevailed up to 12 dpi. Because the M2b MΦ subtype plays an important role in the activation of Th2 cells, this subtype could contribute to establish the microenvironment to favour the replication of parasites in the acute phase of infection. At the end of the blood parasite curve, at 22 and 36 dpi, the M2c and M2d subtypes were the ones that remained and increased more than any other MΦ subtype. These subtypes have properties of tissue remodeling and matrix deposition, and angiogenesis, respectively, which may be related to tissue repair as a consequence of the parasites invading other organs, such as the heart or skeletal muscle after 36 dpi. This also correlated with the increase in VEGF, also observed at 36 dpi, which is involved in cellular infiltration and the repair of tissue damage.

It is interesting to note that there are no previous report in the literature regarding the MΦ subpopulation kinetics in the *T. cruzi* infection model that could help us sustain an interpretation of the temporary function of different M2 subpopulations. Nevertheless, it is worth to consider that *T. cruzi* infection is a highly dynamic process, with parasites coming in and out of cells, and consequently, the behaviour of the M2 subpopulations could reflect the change in the establishment of the parasites in a particular niche or tissue. Without a doubt, these changes deserve further investigation that would help us ¿understand the dynamics of the infection in different cell subpopulations and other target cells.

## 4. Materials and Methods

### 4.1. Animals

Eight- to ten-week-old C57BL/6 and BALB/c female mice were obtained and maintained in the animal facility at CINVESTAV under specific pathogen-free conditions in individually ventilated cages on a 12-h light/dark cycle at 20–22 °C and 40–60% humidity, with access to food and water ad libitum. Animals were acclimated for 1 week prior to experimentation and were sacrificed in a 3% isoflurane chamber (PISA Laboratories, Pachuca, Hgo, Mexico). 

### 4.2. Ethics Statement

All of the experimental procedures were designed to minimize suffering and the number of subjects used. These studies were conducted in accordance with the ethical standards approved and carried out under strict accordance with the guidelines for the Care and Use of Laboratory Animals adopted by the U.S. National Institutes of Health and the Mexican Guideline Regulation of Animal Care and Maintenance (NOM-062-ZOO-1999, 2001) (https://www.gob.mx/senasica/documentos/nom-062-zoo-1999, accessed on 28 December 2015) and the Internal Ethics Committee for the Care and Use of Laboratory Animals (CICUAL). CICUAL-CINVESTAV approved the experimental protocol (Protocol 153-15). 

### 4.3. Trypanosoma cruzi Strain

All of the experiments were conducted using *T. cruzi* blood-derived trypomastigote of the Ninoa strain (MHOM/MX/1994) originally isolated from a patient in Oaxaca (Mexico) [44]. The trypomastigote forms of the *T. cruzi* Mexican strain (TcI group) were maintained by sequential murine passages and collected from the blood of infected mice at the peak of infection (15 days). 

The genetic diversity of *T. cruzi* has been grouped into six genetic subdivisions or “discrete typing units” (DTU), designated as TcI to TcVI [31]. This diversity influences the biological, clinical, immunological, and epidemiological variation of the disease, and is also directly related to the establishment of infection [31,45]. Strains of TcI and TcII are considered the main causative agents of Chagas disease worldwide and especially in South America, where the disease and the two groups are more prevalent [32]. The Mexican *T. cruzi* Ninoa strain belongs to the TcI subtype and has particular biological characteristics, such as patent parasitaemia, non-clinical manifestations, low virulence, and low mortality; a differential immune response is induced in an experimental acute and chronic Chagas disease mouse model [30,32]. When the infection is via oral route, there is a low inflammatory infiltrate in the intestinal mucosa but more than 50% of the infiltrated cells are MΦ [29]. 

### 4.4. Experimental Groups and Infection Challenge

The mice were randomly distributed (n = 5 per group per experiment) and received an intraperitoneal (ip) single inoculation of sterile phosphate-buffered saline (PBS) solution (100 µL) containing 7.5 × 10^3^ parasites. Parasitaemia was evaluated every 4 days (days post-infection; dpi) by counting trypomastigotes in 5 μL of tail blood (1:50 in physiological saline with 0.1% EDTA) and the parasite number per millilitre was determined using a haemocytometer (Hausser Scientific, Horsham, PA, USA) and direct observation under an optical microscope (Olympus Optical Co., Ltd., Tokyo, Japan). The early acute infection was evaluated in two independent experiments. The body weight of the infected and non-infected mice (controls) was recorded at 4-day intervals using a compact portable scale. Upon euthanasia, the heart, spleen, inguinal lymph nodes, liver, lungs, and kidneys were removed from the infected and non-infected mice (controls). We measured them in absolute values and transformed the values to relative weights: as organ (g) to body weight (g) multiplied by 100.

### 4.5. Histopathology

The infected mice were sacrificed 36 and 180 dpi. The control group, non-infected age-matched mice, were sacrificed at the same time as the infected mice. The myocardium was collected and fixed in 4% neutral *p*-formaldehyde and embedded in paraffin. Serial longitudinal sections of 5 µm were stained with haematoxylin and eosin (H&E), or Masson’s trichrome. The preparations were examined to quantify the levels of inflammation and fibrotic lesions in 20 consecutive fields using a light microscope (Nikon). ImageJ software version 1.50i (available from NIH, Bethesda, MD, USA) was used to quantify the fibrotic surfaces in all of the samples.

### 4.6. Single-Cell Preparation from Spleen and Inguinal Lymph Nodes

The inguinal lymph nodes and spleen were collected and digested into single cells suspension under sterile conditions. Briefly, the organs were placed in ice-cold (4 °C) sterile PBS, mechanically disaggregated through a 40 μm cell strainer (BD Falcon), filtered with Roswell Park Memorial Institute (RPMI) medium (Gibco BRL), and centrifuged at 453× *g* for 5 min. Splenocytes were re-suspended in ammonium chloride solution (NH_4_Cl 150 mM, J.T. Baker; KHCO_3_ 10 mM; Na_2_EDTA 0.1 mM, Sigma-Aldrich, Burlington, MA, USA), incubated at 37 °C for 3 min to lysed red blood cells (RBC), and washed twice in PBS. We resuspended the cells in ice-cold PBS.

### 4.7. Cell Isolation from the Peritoneal Cavity

Peritoneal exudate cells (PECs) were obtained by washing the peritoneal cavity of the mice with 8 mL ice-cold PBS plus 3% foetal bovine serum (FBS, Gibco BRL) injected ip. After a soft massage of the peritoneum, the peritoneal cells were recovered and centrifuged at 453× *g* for 5 min at 4 °C. We adjusted the cell density at 10^6^ cells/mL in ice-cold RPMI media. 

### 4.8. Flow Cytometry Staining

Single-cell suspensions of PECs, spleen or lymph nodes were stained with different antibodies and analysed by flow cytometry (FACS). Briefly, 1 × 10^6^ cells/mL were suspended in FACS staining buffer (5% FBS, 0.02% sodium azide in PBS) containing the corresponding antibodies (Ab), at the previously established Ab concentration, for 45 min at 4 °C in the dark. The cells were washed three times with FACS staining solution, fixed with 1% *p*-formaldehyde containing 0.02% sodium azide in PBS, and stored at 4 °C until FACS analysis.

To determine the cell subsets, the cells were stained for extracellular markers with hamster anti-mouse CD3e-PerCP (Cat. 553067; 0.2 mg/mL), rat anti-CD4-PE (Cat. 553049; 0.2 mg/mL), and rat anti-mouse CD8a-Pacific Blue (Cat. 558106; 0.2 mg/mL); these antibodies were purchased from BD-Biosciences Pharmigen (San Diego, CA, USA). Moreover, anti-mouse F4/80-eFluor 450 monoclonal antibody (BM8) (Cat. 48-4801-82; 0.2 mg/mL), anti-mouse CD335-PE-Cyanine 7 (NKp46) (Cat. 25-3351-82; 0.2 mg/mL), and anti-mouse MHC Class II-APC-eFluor 780 (I-A/I-E) monoclonal antibody (M5/114.15.2) (Cat. 47-5321-80; 0.2 mg/mL) were purchased from eBioscience Thermo Fisher Scientific (San Diego, CA, USA). The anti-mouse CD206-FITC (MMR) monoclonal antibody (Cat. 141704; 0.5 mg/mL) was supplied by BioLegend (San Diego, CA, USA).

The cells were concomitantly stained for intracellular content using fixation/permeabilisation solution (Cytofix/Cytoperm, BD Biosciences, San Diego, CA, USA) and anti-mouse CD309-APC (FLK1, VEGF) monoclonal antibody (Avas12a1) (Cat. 17-5821-80; 0.2 mg/mL), anti-mouse latency-associated peptide (LAP, TGFβ) monoclonal antibody-PE-Cyanine7 (TW7-16B4) (Cat. 25-9821-82; 0.2 mg/mL), anti-mouse TNFα-FITC monoclonal antibody (MP6-XT22) (Cat. 11-7321-81; 0.5 mg/mL) and anti-mouse IL-10-PE monoclonal antibody (JES5-16E3) (Cat. 12-7101-82; 0.2 mg/mL); these antibodies were purchased from eBioscience Thermo Fisher Scientific (San Diego, CA, USA). 

### 4.9. FACS Analysis

Stained PECs were acquired (1 × 10^5^ events), as well as splenocyte and lymph node cells (5 × 10^4^ events) using an LSR Fortessa flow cytometer (BD Biosciences, San Diego, CA, USA). The data were analysed (percentage of positively labelled cells) with FlowJo v10.0.7 software (Tree Star, Inc., Ashland, OR, USA). Initially, a double discrimination comparing the side scatter-area (SSC-A) to side scatter-height (SSC-H) and the forward scatter-area (FSC-A) to forward scatter-height (FSC-H) was made to identify single cells and exclude aggregates. Different cell populations were discriminated by morphology cell size (FSC-A) and granularity (SSC-A) and were gated to eliminate red blood cells and debris. We identified the total lymphocytes, helper, cytotoxic, and NK cells as the percentage of cells expressing the markers CD3+, CD4+, CD8+, and CD335+, respectively. 

Global macrophages (MΦ) were identified by F4/80+ (M1) and CD206+ (M2) expression and SSC characteristics. Phenotypification of each MΦ subpopulation was based on the percentage of cells expressing the markers MHC-II+TNFα+ (M1), IL-10+CD206+ (M2a), MHC-II+CD206+ (M2b), IL-10+TGFβ+ (M2c), and IL-10+VEGF+ (M2d) from F4/80+ cells as previously reported [46,47,48] and shown in Figure 7. 

### 4.10. Statistical Analysis

The data show the mean ± standard deviation (SD) of 2 to 3 independent experiments. The Kolmogorov-Smirnov test, Student’s *t*-test or one-way ANOVA with a Bonferroni post hoc were used for comparison between the BALB/c vs. C57BL/6 mice. We considered statistically significant differences between groups at *p*-values lower than or equal to 0.05 (*p* ≤ 0.05). We conducted all of the statistical analyses with GraphPad Prism 7.0 software (GraphPad Software, Inc., San Diego, CA, USA).

## 5. Conclusions

In conclusion, the results presented in this study indicate that the Ninoa strain has developed strategies to regulate the immune response, particularly MΦ, and is a fine regulation that depends on the genetic background of the mouse. They also reveal that, independently from the time taken to resolve the infection; the damage to the heart is equally persistent and could contribute to the heart failure disease observed in Chagas patients decades after the infection. It is worth noting that this is the first time that a fine characterisation of the M2 MΦ populations is presented, and that this characterisation indicates that M2d MΦ are relevant in the induction of heart fibrosis.

## Figures and Tables

**Figure 1 pathogens-10-01444-f001:**
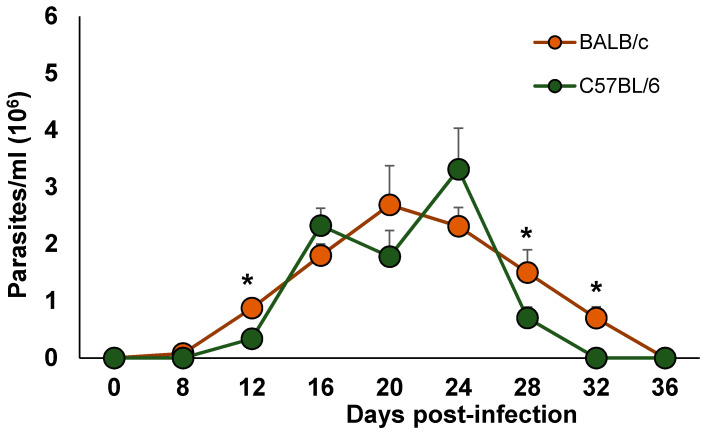
Time-dependent parasite burden in blood. BALB/c and C57BL/6 female mice were inoculated via intraperitoneal (ip) with 7.5 × 10^3^ parasites *T. cruzi* Ninoa strain. Parasite numbers in blood was counted in a Neubauer chamber using light optical microscopy. The results are the mean of three experiments. Mean ± SD, * *p* < 0.05, n = 8, ANOVA post hoc Bonferroni.

**Figure 2 pathogens-10-01444-f002:**
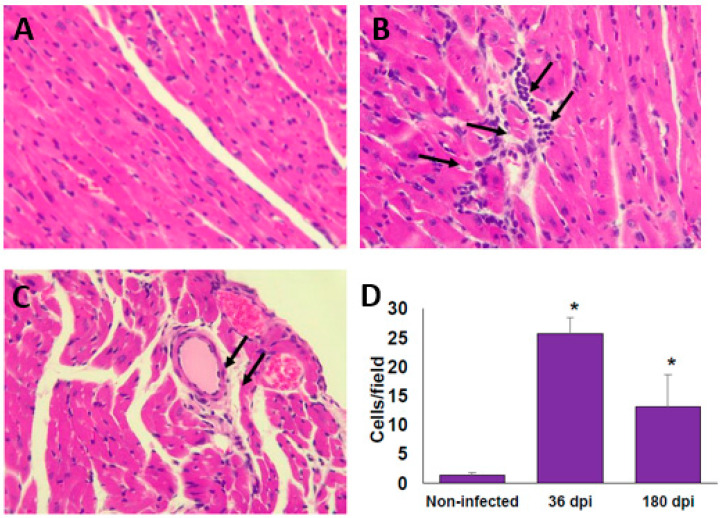
Infiltrated cells in cardiac tissue of BALB/c mice infected with *T. cruzi* Ninoa strain. Mice infected with 7.5 × 10^3^ parasites were sacrificed at 36 and 180 days post-infection (dpi), and hearts were recovered. Tissue was fixed in 4% *p*-formaldehyde and embedded in paraffin. Five μm sections were stained by haematoxylin and eosin (H&E) staining. Non-infected (**A**); 36 dpi (**B**); and 180 dpi (**C**). Microscopy quantification (**D**). Magnification 40X in an optical light microscope. Arrows indicate infiltrating cells. Mean ± SD, n = 8. * *p* < 0.05, ANOVA post hoc Bonferroni vs. non-infected.

**Figure 3 pathogens-10-01444-f003:**
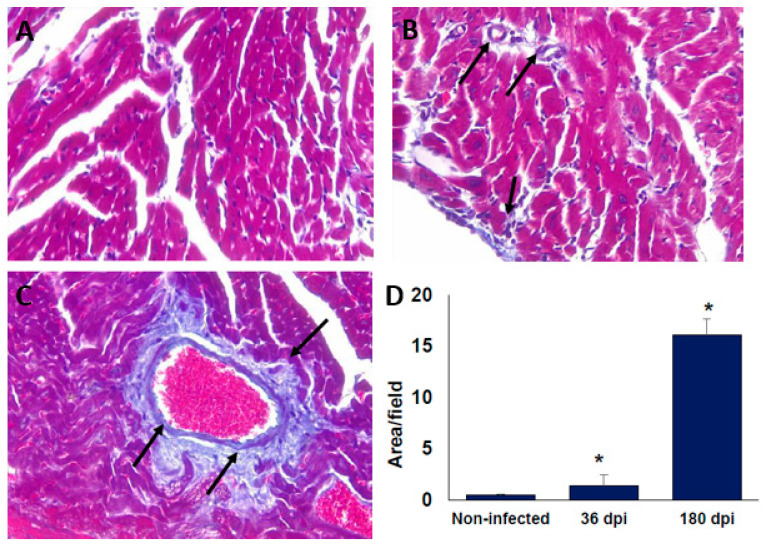
Collagen deposits in cardiac tissue of BALB/c mice infected with *T. cruzi* Ninoa strain. Mice infected with 7.5 × 10^3^ parasites were sacrificed at 36 and 180 days post-infection (dpi), and hearts were recovered. Tissue was fixed in 4% *p*-formaldehyde and embedded in paraffin. Five μm sections were stained with Masson staining. Non-infected (**A**); 36 dpi (**B**); and 180 dpi (**C**). Microscopy quantification (**D**). Magnification 40X in an optical light microscope. Arrows indicate collagen deposits. Mean ± SD, n = 8. * *p* < 0.05, ANOVA post hoc Bonferroni vs. non-infected.

**Figure 4 pathogens-10-01444-f004:**
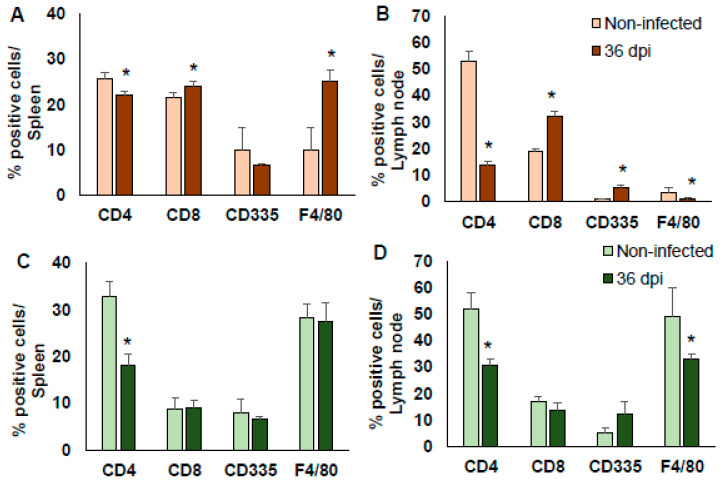
Subpopulations of immune cells post-infection with *T. cruzi* Ninoa strain. Mice infected with 7.5 × 10^3^ parasites were sacrificed at 36 dpi, and their spleens and lymph node cells were stained with specific monoclonal antibodies against CD4+ and CD8+ lymphocytes, NK cells (CD335+), and macrophages (F4/80+). Cells were analysed by flow cytometry. Non-infected mice were used as controls of BALB/c (**A**,**B**) and C57BL/6 mice (**C**,**D**). The bars represent the mean ± SD, n = 5. * *p* < 0.05, ANOVA post hoc Bonferroni vs. non-infected.

**Figure 5 pathogens-10-01444-f005:**
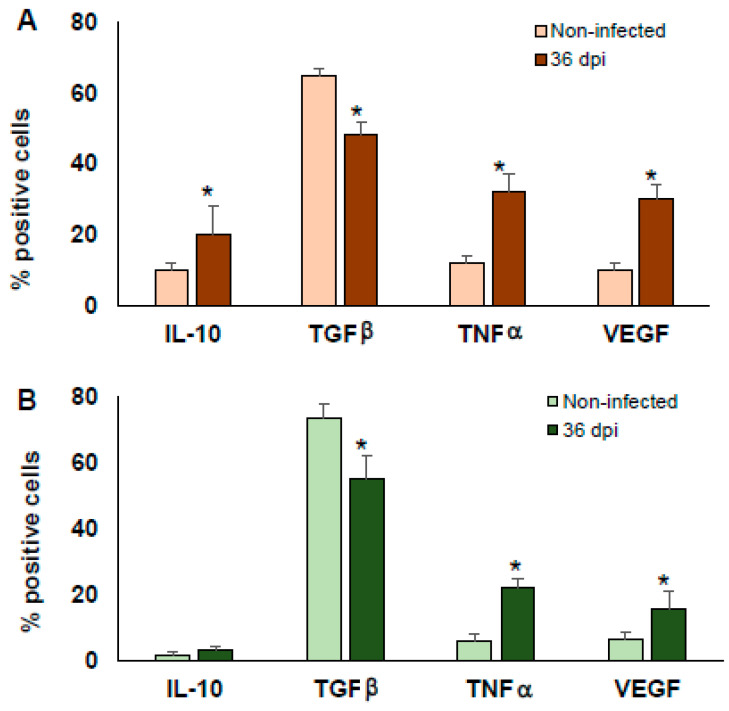
Intracellular cytokines from PECs post-infection with *T. cruzi* Ninoa strain. Mice infected with 7.5 × 10^3^ parasites were sacrificed at 36 dpi, and PECs were stained with specific monoclonal antibodies against F4/80 and intracellular cytokines IL-10, TGFβ, TNFα and VEGF. Cells were analysed by flow cytometry. Non-infected mice were used as controls. BALB/c (**A**) and C57BL/6 mice (**B**). Mean ± SD, n = 5. * *p* < 0.05, ANOVA post hoc Bonferroni vs. non-infected.

**Figure 6 pathogens-10-01444-f006:**
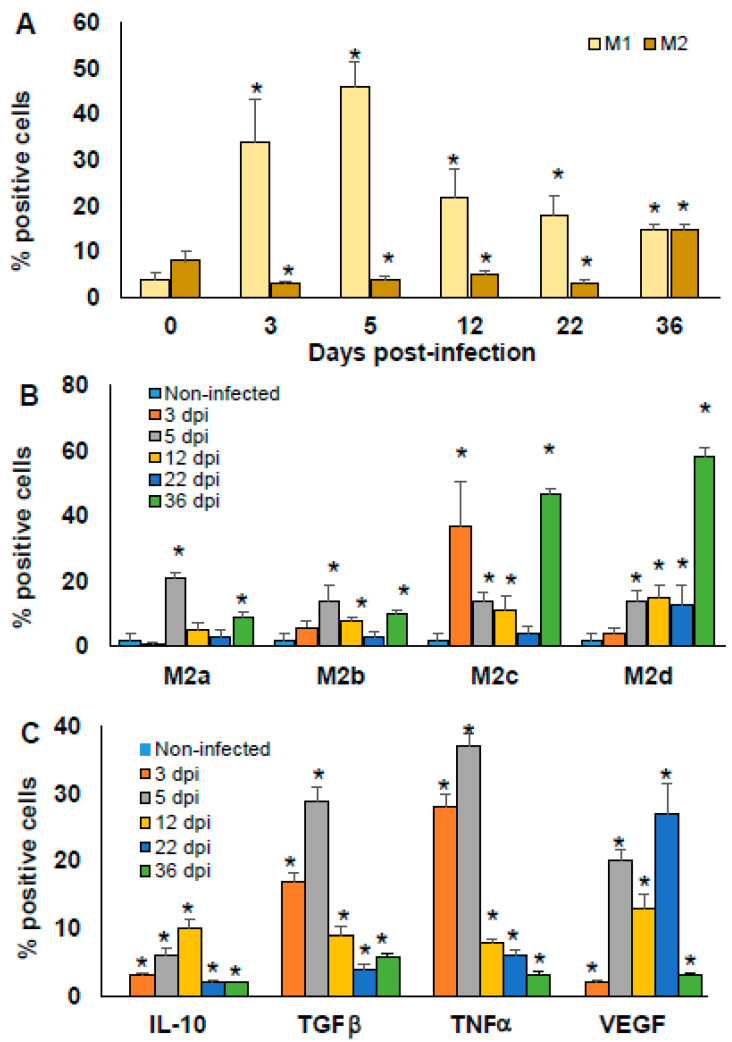
Time-dependent percentage of peritoneal macrophages M1 vs. M2. BALB/c mice infected with 7.5 × 10^3^ parasites *T. cruzi* were sacrificed at 0, 3, 5, 12, 22, and 36 dpi. Adherent PECs were marked with specific monoclonal antibodies against F4/80 and intracellular cytokines IL-10, TGFβ, TNFα, and VEGF. MΦ were analysed by flow cytometry and classified as M1 and M2 phenotype (**A**), or discriminated into M2 MΦ subpopulations M2a, M2b, M2c, and M2d (**B**). Intracellular cytokine production by MΦ was also evaluated (**C**). The bars represent the mean ± SD, n = 5. * *p* < 0.05, ANOVA post hoc Bonferroni vs. non-infected.

**Figure 7 pathogens-10-01444-f007:**
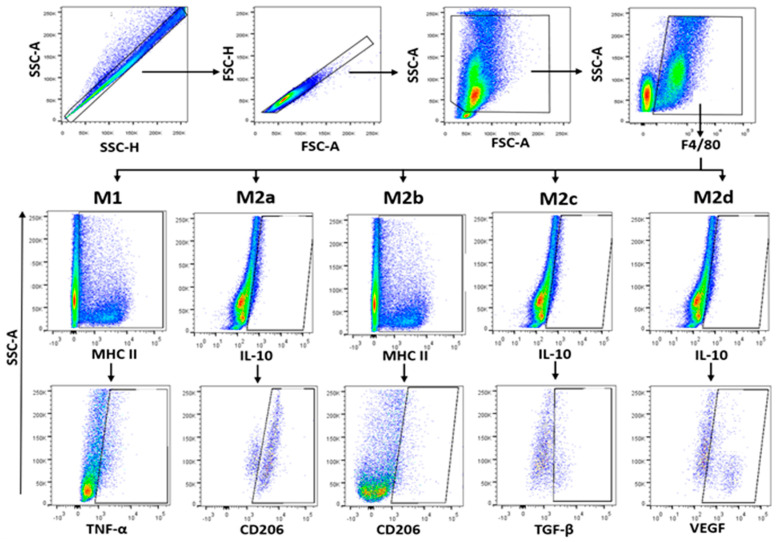
Immunophenotyping strategy for PECs. Freshly isolated PECs were stained as indicated in Section 4.8. Immunophenotyped was performed in 1 × 10^5^ events. We selected single cells to analyse monocytes (F4/80). Further discrimination of M1 and M2 cells was determined by extracellular expression of MHC-II and CD206. Intracellular expression of cytokines was used to determine subsets of M2 cells. Data were analysed in an LSR Fortessa flow cytometer (BD Biosciences) and FlowJo v10.07 software.

**Table 1 pathogens-10-01444-t001:** Relative organs weight from non-infected and infected BALB/c and C57BL/6 mice.

	BALB/c		C57BL/6	
Organ ^1^	Non-infected	36 dpi ^2^	Non infected	36 dpi
Spleen	0.096 ± 0.01	2.97 ± 0.21 *	0.098 ± 0.03	2.40 ± 0.50 *
Liver	0.87 ± 0.13	7.32 ± 0.19 *	0.97 ± 0.15	6.30 ± 0.20 *
Heart	0.126 ± 0.01	0.66 ± 0.06	0.14 ± 0.03	0.60 ± 0.10
Lymph node	0.092 ± 0.14	0.06 ± 0.00 *	0.098 ± 0.24	0.06 ± 0.01 *
Body weight (g)	17.38 ± 0.94	17.56 ± 0.15	18.43 ± 0.29	18.70 ± 0.91

^1^ Relative weight of organs. ^2^ dpi; days post-infection. Mean ± SD (n = 5). * *p* < 0.05 ANOVA post hoc Bonferroni vs. non-infected.

## Data Availability

The data can be made available upon request to the authors.

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
