# Peer review of "Characterisation of Macrophage Polarisation in Mice Infected with Ninoa Strain of Trypanosoma cruzi"

_pathogens, 2021, doi:10.3390/pathogens10111444_

Round 1

Reviewer 1 Report

The MS entitled "Characterisation of macrophage polarisation in mice infected with Ninoa strain of Trypanosoma cruzi" by Dunia M. Medina-Buelvas, Miriam Rodríguez-Sosa, and Libia Vega, is an interesting work that  made a correlation between the evolution of the infection with T. cruzi DTU-I (Mexican strain) in two different strains of mice (Balbc and C57) and the macrophage polarization. Although some conclusions are sound and supported by the data other important aspects should be completed/improved before the final acceptance of the work. 

Major points:

  • It is documented that mice male are more susceptible to T. cruzi infection than female (Micucci et al, 2010 Rev. Fac. Cien. MEd. Univ. Nac. Córdoba 2010;67(2):73-6.;  Mena-Marin et al, 2012 Bol Mal Salud Amb vol 52 n2 Maracay Ago 2012). Why do you select female? Explanation for this selection should be included in the introduction, Moreover; absence of mortality of mice could be for the use of female besides inoculum.
  • What conclusions are obtained from Fig 4? Due to this study was done in the spleen and lymph nodes, which could be the relationship with the cytokine profile of PECs? Most interesting data is to know the cellular content of peritoneal exudate and then relate with the cytokine profile.
  • In the cytokine profile showed in Fig 5, what can be said about the immune response of Balbc and C57 concerning the pro or anti-inflammatory responses? Although some ideas were mentioned in the discussion, a more deep analysis of these data including comparison with recent bibliography should be added. For example; why Il-10 present in PECs from Balb C mice coexists with TNF-a?  M
  • Why only the macrophage polarization of Balbc mice was studied? What about the resistant C57 strain? Authors set up that interaction of T. cruzi with macrophages and the polarization of them is important to know the type of response (inflammatory or not) that will be produced during acute infection and that will be involved in the outcome of disease. Eventhough, they did not show the profile of M2 response in C57 mice. This is needed to verify possible differences in the immune response of a resistant and susceptible strain of mice and to complete the idea about of the influence of the genetic background in the infection.

Minor point:

Page 8, line 263: should be M2d instead M2c

Author Response

Reviewer 1

1.- a) It is documented that mice male are more susceptible to T. cruzi infection than female (Micucci et al., 2010 Rev Fac Cien Med Univ Nac Córdoba, 2010;67(2):73-6; Mena-Marin et al., 2012 Bol Mal Salud Amb, vol 52, n2, Maracay, Ago 2012). b) Why do you select female? Explanation for this selection should be included in the introduction, c) Moreover; absence of mortality of mice could be for the use of female besides inoculum.

Response: Thank you very much for these proper observations, we have incorporated the information through manuscript.

  1. a) You are right, the male are more susceptible than female mice to T. cruzi infection. We have added this information in the introduction with the suggested citations, page 2 lines 54-56.
  2. b) At the beginning of this study, we infected male and female mice, we did not observe differences in mortality or in clinical manifestations between the sexes, possibly due to the reduced number of parasites that were inoculated. Based on this observation, and due to availability of female mice we used these. Here we attach a figure showing parasitemia in males and females. Although the males had more parasites at 24- and 28-days post-infection than the females, parasitemia in blood was resolved at the same days. Importantly, mortality was not observed in either sex.

Future experiments are required to compare the differences between males and females; however, this objective is out of scope for this paper.

  1. c) We agree with you regarding that the absence of mortality in mice could be due to the use of females besides inoculum. We have incorporated this information in the discussion section. Page 10, lines 310-317.

  1. a) What conclusions are obtained from Fig. 4? b) Due to this study was done in the spleen and lymph nodes, which could be the relationship with the cytokine profile of PECs? c) Most interesting data is to know the cellular content of peritoneal exudate and then relate with the cytokine profile.

Response: Thank you for the observation.

  1. a) We have made the change and the new sentence reads as follows “Analysis at 36 dpi of the proportion of some immune cells showed no difference in the regulation of TD4+ cells between BALB/c and C57BL/6 mice, in both cases they decreased. This behavior was similar that previously reported by Espinoza et al. which a significant de-crease in CD4+ T cell activation was observed in the acute phase of infection with T. cruzi strain Ninoa [28]. Whereas the proportion of TCD8 + cells in spleen and lump nodes was significantly increased in BALB/c mice but not in C57BL/6 mice. Therefore, BALB/c mice developed a mild inflammatory process characterized by a slight increase in the expression of TCD8 +, NK CD335 + and with a considerable increase in MΦ F4/80 + in the spleen, whereas C57BL/6 did not. This supports that Ninoa strain differentially modulates the immune response (even with small differences) depending on the genetic background of the Th1 or Th2 mouse [30]”. We have incorporated this information in the discussion section. Page 11, lines 331-340.
  2. b) We discussed this in the discussion section, page 11, lines 342-352.
  3. c) As we saw a significant difference in the percentage of macrophages in the spleen and lymph nodes between BALB/c vs C57BL/6 mice, we focused on analyzing the production of cytokines associated with the infection in this cell population.

Furthermore, the percentage of negative F4/80 PECs (other cell types, not macrophages) had low production of the cytokines analyzed. We have incorporated this information in the results section. We included supplementary figure 1 where we analyzed intracellular cytokines in F4/80 negative PECs. Page 6, lines 189-191.

Due to this, we no longer analyzed other populations. However, it is a good observation for future experiments. In this article, this objective is out of scope.

  1. In the cytokine profile showed in Fig. 5, what can be said about the immune response of Balb/c and C57 concerning the pro- or anti-inflammatory responses? Although some ideas were mentioned in the discussion, a more deep analysis of these data including comparison with recent bibliography should be added. For example, why IL-10 present in PECs from Balb/c mice coexists with TNF-a?

Response: Thank you for you recommendation.

We have added information in the discussion section according to your suggestion. Page 11, lines 351-362.

Related to the coexistence of IL-10 and TNF-a in the PECs of BALB/c mice, we believe that it is related to the genetic background of the mouse, while BALB/c can produce IL-10, C57BL/6 mice are not. Interestingly, Huynh L., et al. (1) reports that the presence of TNF-a does not negatively regulate IL-10 production in human macrophages. In contrast, TNF-a can induce IL-10 production (2). But, to our knowledge, this finding was not followed up and the regulation of autocrine IL-10 production induced by TNF and subsequent signaling, gene induction and function has not been investigated. On the other hand, these results could indicate a mixed profile and to the coexisting of different polarized or activated cells.

We think, it is a very interesting topic to study in the future.  

  1. Why only the macrophage polarization of Balb/c mice was studied? What about the resistant C57 strain? Authors set up that interaction of T. cruzi with macrophages and their polarization is important to know the type of response (inflammatory or not) that will be produced during acute infection and that will be involved in the outcome of disease. Even though, they did not show the profile of M2 response in C57 mice. This is needed to verify possible differences in the immune response of a resistant and susceptible strain of mice and to complete the idea about of the influence of the genetic background in the infection.

Response: Thank you for this important observation.

Most of the evidence for polarization of M1-M2 macrophages and their participation in T. cruzi infection has been made in human cells (3), so the participation of this subpopulation of macrophages is not in doubt. However, based on our experience, and in other previously published works with infections of other parasites (4, 5) it has been observed that C57BL/6 mice hardly develop M2 macrophages, so we gave priority to the analysis of M2 in infected BALB /c mice.

  1. Page 8, line 263: should be M2d instead M2c.

Response: This has been corrected. Page 7, line 247.

Reviewer 2 Report

The MS entitled “Characterisation of macrophage polarisation in mice infected with Ninoa strain of Trypanosoma cruzi” submitted to Pathogens is a good descriptive article of a T.cruzi strain which is not fully known. Despite is an interesting article I think there is some aspects to improve.

-In line 43 it says literally “After T. cruzi acquisition, the parasite infects blood cells as monocytes and macro-43 phages (MΦ) in which the parasites proliferate” it seems that T.cruzi always entries to the host directly to the blood, but it is not true. Authors should change the redaction of this idea

- I think there are very little differences in the effects produced by T.cruzi infection in balb/c when it is compared to C57BL6, do you have any explanation?. Authors say that parasite burden in c57BL/6 is significant lower at 12,28 and 32 dpi, but is higher at 16 and 24 dpi. Actually, the differences that the authors present result showing something similar to a tolerance, but it is not a resistance.

- In figures 2 and 3, it should be interesting to include some marks in order to point infiltrating cells and collagen deposits. Have you studied which cells are infiltrating heart tissue? Maybe it will be different in balb/c and C57BL6

-  Could the authors explain which is the role that can VEGF and angiogenesis in T.cruzi infection? I think it may be interesting to explain it in the introduction

- The paragraph which starts in line 245 and finishes in line 253 will fit better in introduction than in results. Also in this paragraph it will be interesting that the authors explain what is the “encapsulation of parasites”, regarding T.cruzi is an intracellular parasite which can replicate inside immune cells.

- Why did the authors use F4/80, MHC-II and TNF-α as a markers of M1 instead of CD80 or CD86, which are the most usual markers? In the MS the authors cite another paper which was published by themselves, but in the paper they not cite any other paper.

- In figure 6 is difficult to understand the increase of M2a, M2b and M2c, followed by a decrease, followed by a new increase of the populations, do the authors have an explanation for this behavior?

- Also in figure 6 there is a big increase of M2d 36 dpi which is not in correlation with an increment in VEGF expression 36dpi, do the authors have an explanation?

Author Response

Reviewer 2

  1. Line 43 it says literally “After T. cruzi acquisition, the parasite infects blood cells as monocytes and macrophages (MΦ) in which the parasites proliferate” it seems that T. cruzi always entries to the host directly to the blood, but it is not true. Authors should change the redaction of this idea.

Response: We agree with this comment, thank you for the observation.

We have changed the redaction of the phrase indicating the process once the parasites reach the bloodstream circulation. Page 2, lines 42-43.

  1. a) I think there are very little differences in the effects produced by T. cruzi infection in balb/c when it is compared to C57BL6, do you have any explanation? b) Authors say that parasite burden in c57BL/6 is significant lower at 12, 28 and 32 dpi, but is higher at 16 and 24 dpi. The differences that the authors present result showing something similar to a tolerance, but it is not a resistance.

Response: Thank you, we have incorporated information about these commentaries in discussion section.

  1. The small differences in the effects produced by T. cruzi infection between BALB/c and C57BL /6 could be explained by the fact that different strains of T. cruzi have different infectivity potentials (6). In this sense, the Tc1 genotype, to which the Ninoa strain belongs, has a high intraspecific genetic and phenotypic diversity (7), which is determinant to recognize the surface molecules of the parasite by immune cells (8). These characteristics could explain how the strain of T. cruzi Ninoa exhibits more modulating behaviors of the immune response compared to other strains of T. cruzi (7). Thus, it is possible that the Ninoa strain could manipulate immune cells to produce a tolerogenic cytokine profile regardless of the genetic background of the mouse. Page 11, lines 359-367.
  2. You are right, the small differences in parasites burden between C57BL/6 and BALB/c infected mice suggest tolerance, not resistance. We have change in the discussion section. Page 11, line 342.

  1. In figures 2 and 3, it should be interesting to include some marks in order to point infiltrating cells and collagen deposits. Have you studied which cells are infiltrating heart tissue? Maybe it will be different in balb/c and C57BL6.

Response: Thank you for the recommendation.

We have included arrows indicating infiltrating cells in figure 2 and collagen deposits in figure 3.

  1. Could the authors explain which is the role of VEGF in angiogenesis in T. cruzi infection? I think it may be interesting to explain it in the introduction.

Response: Thank you for this observation. In fact, it is interesting to note that relationship.

Although we cannot find a proper explanation regarding this particular relationship between angiogenesis and the infection of T. cruzi, there are some evidence indicating that VEGF could be related to tissue remodelling. Thus, having an indirect relationship with the infection per se. We have included some information regarding the role of macrophage subtypes in the control of the infection. Page 2, lines 72-75.

  1. The paragraph starting in line 245 and finishing in line 253 will fit better in introduction than in results. Also in this paragraph, it will be interesting that the authors explain what is the “encapsulation of parasites” regarding T. cruzi is an intracellular parasite which can replicate inside immune cells.

Response: Thank you for your proper observation.

We have moved the paragraph to introduction section; page 2, lines 66-75.

Regarding to “encapsulation of parasites”. We apologize for not being clear, encapsulation refers to extracellular parasites (such as filarial nematode Onchocerca volvulus, larval Anguillicoloides crassus, etc.), it does not happen in T. cruzi. We have adding “encapsulation of extracellular parasites” in order to be clear. Line 73.

  1. Why did the authors use F4/80, MHC-II and TNF-α as markers of M1 instead of CD80 or CD86, which are the most usual markers? In the MS the authors cite another paper which was published by themselves, but in the paper they not cite any other paper.

Response: Thank you for pointing this out.

The characterization of macrophage polarization requires a multi-parametric approach to analysis, including surface markers and cytokines that reflect its functionality. We used the surface marker of mouse macrophage F4/80+ expressed on most resident tissue macrophages and tightly regulated according to the physiological status of these cells, being a unique marker of murine macrophages more specific for evaluating peritoneal macrophages (9). In mice, F4/80+ and MHC-II are markers useful, easy, and faster for FACS determination of peritoneal macrophages allowing the identification of subsets according to their morphology as Ghosn et al., (10), where the large and small peritoneal macrophages that express differential levels of these molecules. Also, M1 MΦ has a high production of pro-inflammatory cytokines as TNF-α being a good marker to determine macrophages towards this phenotype (11). An example of this was the study carried out by Haloul et al. in 2019 (12), that used a similar scheme of markers that we used for our study to identify M1 population.

We have added a reference for the selection of markers in the Materials and Methods section. Page 15, line 554.

  1. In figure 6 is difficult to understand the increase of M2a, M2b and M2c, followed by a decrease, followed by a new increase of the populations, do the authors have an explanation for this behaviour?

Response: There is no previous report in the literature regarding the MΦ subpopulation kinetics in the T. cruzi infection model that could help us sustain an interpretation of the temporary function of different M2 subpopulations. Nevertheless, it is worth to consider that T. cruzi infection is a highly dynamic process, with parasites coming in and out of cells, and consequently, the behaviour of the M2 subpopulations could reflect the change in the establishment of the parasites in a particular niche or tissue. We have included this reflection in the discussion section, page 12, lines 422-429.

  1. Also in figure 6 there is a big increase of M2d 36 dpi, which is not in correlated with an increment in VEGF expression 36 dpi, do the authors have an explanation?

Response: Thank you for the observation.

The reviewer is right; the increase in M2d does not appear to correlate with the amount of cells producing VEGF at the same time point. This is true for the other cytokines evaluated, not only VEGF. Thus, we consider that this behavior could be due to a delayed polarization of the cell subpopulations that comes after the increase in the cytokines.

Round 2

Reviewer 1 Report

Authors have responded the questions adequately and completed the introduction and discussion with clarifications and relevant bibliography as recommended. I recommend the acceptation of the MS of Medina-Buelvas DM et al in the present form. 

Reviewer 2 Report

In my opinion, the manuscript with these modifications is ready for publication.